# Obesity Affects the Proliferative Potential of Equine Endometrial Progenitor Cells and Modulates Their Molecular Phenotype Associated with Mitochondrial Metabolism

**DOI:** 10.3390/cells11091437

**Published:** 2022-04-24

**Authors:** Agnieszka Smieszek, Klaudia Marcinkowska, Ariadna Pielok, Mateusz Sikora, Lukas Valihrach, Elaine Carnevale, Krzysztof Marycz

**Affiliations:** 1Department of Experimental Biology, Faculty of Biology and Animal Science, University of Environmental and Life Sciences, Norwida 27B St., 50-375 Wroclaw, Poland; klaudia.marcinkowska@upwr.edu.pl (K.M.); ariadna.pielok@upwr.edu.pl (A.P.); mateusz.sikora@upwr.edu.pl (M.S.); krzysztof.marycz@upwr.edu.pl (K.M.); 2Laboratory of Gene Expression, Institute of Biotechnology CAS, Biocev, 25250 Vestec, Czech Republic; lukas.valihrach@ibt.cas.cz; 3Equine Reproduction Laboratory, Department of Biomedical Sciences, Colorado State University, Fort Colins, CO 80523-1693, USA; elaine.carnevale@colostate.edu; 4International Institute of Translational Medicine, Jesionowa 11 St., 55-124 Malin, Poland

**Keywords:** obesity, endometrial progenitor cells, cellular metabolism, self-renewal potential

## Abstract

The study aimed to investigate the influence of obesity on cellular features of equine endometrial progenitor cells (Eca EPCs), including viability, proliferation capacity, mitochondrial metabolism, and oxidative homeostasis. Eca EPCs derived from non-obese (non-OB) and obese (OB) mares were characterized by cellular phenotype and multipotency. Obesity-induced changes in the activity of Eca EPCs include the decline of their proliferative activity, clonogenic potential, mitochondrial metabolism, and enhanced oxidative stress. Eca EPCs isolated from obese mares were characterized by an increased occurrence of early apoptosis, loss of mitochondrial dynamics, and senescence-associated phenotype. Attenuated metabolism of Eca EPCs OB was related to increased expression of pro-apoptotic markers (CASP9, BAX, P53, P21), enhanced expression of OPN, PI3K, and AKT, simultaneously with decreased signaling stabilizing cellular homeostasis (including mitofusin, SIRT1, FOXP3). Obesity alters functional features and the self-renewal potential of endometrial progenitor cells. The impaired cytophysiology of progenitor cells from obese endometrium predicts lower regenerative capacity if used as autologous transplants.

## 1. Introduction

As is widely known, obesity contributes to an increased risk of cardiovascular disease, type 2 diabetes, and stroke. However, there is also a growing body of evidence supporting the negative influence of obesity on reproduction, including infertility, low quality of oocytes, and increased early pregnancy loss [1]. Additionally, obesity reduces the success of attaining pregnancy, even when accompanied by assisted reproductive technology. It was also shown that obesity is strongly associated with the development of endometrial cancer. The endometrial environment has an essential role in obesity-related reproductive impairments, but molecular mechanisms by which obesity contributes to the degeneration of the endometrium niche are still being investigated. However, mitochondrial dysfunction is highlighted as a significant link between cellular metabolic imbalance and reproductive problems [2].

In this study, we have incorporated the equine model to study the effect of obesity on the cytophysiology of endometrial progenitor cells.

From studies conducted in Europe, North America, Australia, and New Zealand, 2 to 72% of horses are considered overweight, and 1 to 19% of animals are obese [3,4]. Obesity contributes to developing metabolic disorders, such as insulin resistance, equine metabolic syndrome (EMS), and laminitis—all conditions accompanied by inflammation. Infertility and altered reproductive function of mares are also consequences of obesity [4,5]. Maternal obesity also influences foal metabolism and increases the occurrence of low-grade inflammation and osteochondrosis in yearlings [3]. The effect of obesity on the equine uterus has not been well defined, although gene expression in the endometrium was altered even after short-term obesity, showing a pro-inflammatory profile with increased expression of TNFα and IL1β [6].

Excessive accumulation and expansion of adipose tissue affect the endogenous self-renewal capacity of the organism [7]. Our previous studies have demonstrated that equine metabolic syndrome affects the regenerative potential of adipose-derived stromal/stem cells (ASCs) [8,9,10,11]. Progenitor cells, isolated from horses with metabolic syndrome, have limited clinical potential compared to healthy subjects because of reduced proliferation capacity, disturbed cellular homeostasis, and hampered multipotency. The physiological status and age of the horse affect the metabolism of progenitor cells. Aging and equine metabolic syndrome reduce the self-renewal potential of progenitor cells, significantly impairing mitochondrial function while increasing autophagy, oxidative stress, and endoplasmic stress [8,9,10]. The lowered regenerative features of progenitor cells, resulting from obesity-induced cellular damage, notably limit therapies based on autologous transplants [12].

The biology of equine multipotent stromal/stem cells (MSCs) isolated from adipose tissue and bone marrow is well described. These cells are commonly used to treat equine locomotive disorders [12,13]. However, since MSCs keep an epigenetic memory specific to the tissue of origin, other adult tissues are being explored as potential sources for progenitor cells [14]. Recently, attention has been given to the cytophysiology of progenitor cells originating from the endometrium [14,15,16]. The endometrium has increased regenerative potential, which is needed for remodeling during the estrous cycle. Progenitor cells from the uterus can be harvested by a biopsy, which is a nonsurgical procedure without the need for anesthesia in mares. The endometrium provides an attractive source of easily obtained and highly proliferating cells [14,15,17]. 

Cells isolated from equine endometrial tissue were previously characterized as a multipotent population with enhanced self-renewal properties [15]. Uterine pathologies, such as post-breeding endometritis and endometriosis, lead to a substantial economic burden for the equine industry [18]. Thus, therapies based on the application of endometrial progenitor cells are appealing.

The main objective of this study was to evaluate the effects of obesity on basic cytophysiological features of equine endometrial progenitor cells (Eca EPCs), including viability and self-renewal potential, as well as mitochondrial metabolism. We compared proliferation capacity, mitochondrial dynamics, and molecular phenotypes of EPCs obtained from non-obese (non-OB) and obese mares (OB).

## 2. Materials and Methods

### 2.1. Harvesting and Isolation of Equine Endometrial Progenitor Cells (Eca EPCs) from Obese and Non-Obese Mares

Tissue samples were obtained post-mortem from non-obese (*n* = 4) and obese (*n* = 4) mares during anestrus. Uteri were collected from a local slaughterhouse at Rawicz (Poland) in November and December of 2019 and 2020, and samples were isolated from the uterine body. The mean age of non-obese mares was 9.5 ± 2, while obese mares were 9 ± 1. The obesity of mares was determined based on body condition score (BCS), using the protocol established previously [10,19] and a system developed by Henneke et al. [20]. BSC is expressed by numerical values for fat deposition ranging from 1 (poor) to 9 (extremely obese). The horses in the non-obese group had BSC = 6.5 ± 0.289 (mean + SEM), while obese mares were characterized by BCS = 8.75 ± 0.250 (mean + SEM).

The tissue fragments (~4 g) were washed three times using Hank’s balanced salt solution (HBSS, Sigma Aldrich/Merck, Poznan, Poland) supplemented with 1% of penicillin/streptomycin mix (P/S, Sigma Aldrich/Merck, Poznan, Poland). The endometrium was dissociated from myometrium, cut into small fragments, and placed into an enzymatic solution. The solution consisted of collagenase type I (1 mg/mL) prepared in Dulbecco’s modified Eagle’s medium/F12 (DMEM/F12, Sigma Aldrich/Merck, Poznan, Poland). The tissue fragments were digested for 40 min in CO_2_ incubator at 37 °C with periodic vortexing. After incubation, the homogenates were filtered using a sterile 70-μm cell strainer (Greiner Bio-One, Biokom, Janki, Poland) to discard the undigested tissue fragments. Further, the suspension of cells was centrifuged at 300× *g* for 10 min. Obtained pellets were washed in HBSS and then resuspended in DMEM/F12 medium supplemented with 15% of fetal bovine serum (FBS, Sigma Aldrich/Merck, Poznan, Poland) and 1% P/S solution, hereinafter referred to as a complete growth medium for endometrial stromal cells (CGM_EPC_). The cells were transferred into a T-75 culture flask (Nunc, Biokom, Janki, Poland). Both primary and subsequently passaged cultures were cultured at constant conditions in CO_2_ incubator at 37 °C and maintained 95% humidity. The CGM_EPC_ was changed every 2 days. The culture was passaged when reaching 90% confluence using a trypsin solution (StableCell Trypsin, Sigma Aldrich/Merck, Poznan, Poland). Cells used for the experiment were pulled at passage 3 (p3).

### 2.2. Phenotype and Multipotency of Eca EPCs

The phenotype of cells was determined based on mRNA expression of (i) the mesenchymal markers, including CD29, CD44, CD90, CD105; (ii) hematopoietic markers, i.e., CD34 and CD45; (iii) perivascular markers, including CD146 and NG2; (iv) an epithelial marker, i.e., mucin-1 and (v) smooth muscle markers, including actin alpha 1 (ACTA2), calponin 1 (CNN1) and myosin heavy chain 11 (MHY11).

The expression of selected genes was determined using the RT-qPCR protocol described in Section 2.6. The sequences of oligonucleotides were published by Rink et al. [15] and were fully characterized in Appendix A. Amplified PCR products were separated in 2% agarose gel and stained with SYBR™ Safe DNA Gel Stain Thermo Fisher Scientific, Warsaw, Poland. The molecular weight of the obtained products was compared to DNA Ladder (M50pz, Blirt DNA, Gdansk, Poland).

Immunocytochemical staining aimed at detecting CD44, CD45, CD73, CD90, and CD105 surface markers is described in Section 2.8, while antibodies used for analysis are listed in Appendix A.

To verify the multipotency, Eca EPCs were cultured under osteogenic, chondrogenic, and adipogenic conditions. For this purpose, cells (p3) were seeded into 6-well plates at inoculum 1 × 10^5^ cells per well. The differentiation was induced using commercially available kits (StemPro^®^, Thermo Fisher Scientific, Warsaw, Poland) after 48 h of culture when cells reached 90% confluency. The effectiveness of tissue-specific differentiation was evaluated after 21 days in osteogenic cultures and after 16 days in chondrogenic and adipogenic cultures. Following differentiation, the extracellular matrix was stained accordingly to well-established protocols [21]. Osteogenesis in vitro was verified with Alizarin Red staining. The chondrogenic nodules were detected with Safranin O. In contrast, lipid droplets in adipogenic cultures were detected after Oil Red O staining. Documentation of cultures was performed using an inverted microscope (Axio Observer A.1, Zeiss, Oberkochen, Germany) with Power Shot digital camera (Canon, Woodhatch, UK). Images were processed and analyzed, as was previously described [22].

### 2.3. Evaluation of Proliferation and Migratory Capacity of Cells

Proliferation activity of cells was evaluated based on multiple parameters established in a clonogenic and wound healing assay, MTS assay, and cell cycle analysis. The protocols were used previously for progenitor cells of different origins and were published in detail [11,23,24]. In this experiment, the efficiency of colony-forming unit (CFU-E) occurrence was evaluated after 12 days of culture using the protocol described by Rink et al. [15]. Cells were inoculated in 6-well plates at a density equal 10 cells/cm^2^ in 2 mL of CGM_EPC_. After the test, cultures were fixed in 4% of paraformaldehyde (PFA) for 30 min, subsequently washed using HBSS, and stained with 2% pararosaniline solution (Sigma Aldrich/Merck, Poznan, Poland) for 5 min before being rewashed with HBSS. Colonies were defined as clusters formed by more than 50 cells.

For the wound healing assay, cultures were maintained at a high confluence, reaching 90%. The scratch was made with a 200-μL pipette tip, and cultures were propagated for an additional 24 h. After that, cultures were fixed with 4% PFA and stained with the pararosaniline solution. Cultures were documented using a Canon PowerShot digital camera (Canon, Woodhatch, UK).

The population doubling time (PDT) was determined based on the growth curve of Eca EPCs, established by a routine counting of cells during the passage. The cells were counted using the Muse^®^ Count and Viability Kit (Luminex/Merck, Poznan, Poland) according to the manufacturer’s protocol and analyzed using Muse Cell Analyser (Sigma-Aldrich/Merck, Poznan, Poland). Obtained data were used to determine PDT with an algorithm published by Heuer et al. [25] and Cell Calculator [26].

The proliferation determined based on metabolic activity was measured using MTS (Cell Proliferation, Colorimetric, Abcam, Cambridge, UK). For this purpose, Eca EPCs were seeded in 96-well plates at a density equal to 5 × 10^3^ per well in 200 μL CGM_EPC_. After 24 h of cells inoculation, 20 μL of MTS solution was added per well. The test included blank control. Cultures were incubated with the dye for 2 h at 37 °C in a CO_2_ incubator. Following incubation, absorbance was measured spectrophotometrically with a plate reader (Epoch Biotek, Biokom, Janki, Poland) at a wavelength of 490 nm. MTS assay was performed after 24, 48, and 72 h of Eca EPCs culturing. 

The cell cycle was determined using Muse Cell Cycle Kit (Luminex/Merck, Poznan, Poland) using the manufacturer’s protocol. The analysis was performed using Muse Cell Analyser (Sigma-Aldrich/Merck, Poznan, Poland), and for each assay, 0.5 × 10^6^ Eca EPCs were tested. 

The proliferation activity of Eca EPCs from non-obese and obese mares was also evaluated based on immunocytochemical staining of Ki67 marker and miRNA levels. A transwell system with a pore size of 8 µm (Corning, Biokom, Janki, Poland) was used to test the motility capabilities of Eca EPCs. Cells were inoculated into the upper chamber at a density equal to 0.2 × 10^5^ in a 300 μL of serum-free culture medium (DMEM/F12 supplemented 1% P/S). The CGM_EPC_ was added to the lower compartment of the transwell system. The cells were stained with pararoseline solution, as described above. Chambers were evaluated under an inverted microscope (Leica DMi1) with an attached Digital Camera MC170 and Leica Application Suite (LAS) software (the equipment and software derived from Leica Microsystems, KAWA.SKA Sp. z o.o., Zalesie Gorne, Poland).

### 2.4. The Evaluation of Eca EPC Growth Pattern, Morphology, and Ultrastructure

The growth pattern and morphology of cultures were evaluated using an inverted microscope (Leica DMi1) equipped with a camera MC170 (Leica Microsystems, KAWA.SKA Sp. z o.o., Zalesie Gorne, Poland). Cell condition was also determined from images obtained after senescence-associated beta-galactosidase staining. The ultrastructure of Eca EPCs was analyzed using a confocal microscope (Leica TCS SPE, Leica Microsystems, KAWA.SKA Sp. z o.o., Zalesie Gorne, Poland) based on localization and distribution of nuclei, cytoskeleton, and mitochondrial network. The mitochondria were stained in vital cultures with MitoRed dye (Sigma-Aldrich/Merck, Poznan, Poland) prepared in CGM_EPC_ (1:1000). Subsequently, cultures were incubated with the dye in CO_2_ incubator at 37 °C for 30 min. Next, cells were fixed with 4% paraformaldehyde (PFA, Sigma-Aldrich/Merck, Poznan, Poland) for 30 min at room temperature and rinsed three times with HBSS. Following fixation, the actin cytoskeleton was imaged using Phalloidin-Atto 488 (Sigma-Aldrich/Merck, Poznan, Poland). For this purpose, cells were permeabilized with 0.2% Tween 20 and prepared in HBSS for 20 min at room temperature. After permeabilization, cells were stained with phalloidin solution dissolved in HBSS at a concentration of 1:800. The staining lasted 30 min and was performed at 37 °C in the dark. Cells nuclei were counterstained using 4′,6-Diamidine-2′-phenylindole dihydrochloride (DAPI) mounting medium (ProLong™ Diamond Antifade Mountant with DAPI, Thermo Fisher Scientific, Warsaw, Poland). Preparations were observed using the confocal microscope under magnification 630× and processed with Fiji software (ImageJ 1.52n, Wayne Rasband, National Institute of Health, Bethesda, MD, USA). The three-dimensional reconstructions of the mitochondrial network were performed using Leica Application Suite X (version 3.5.2.18963, Leica Microsystems CMS GmbH). For this purpose, confocal images were processed using the “3D viewer” option. Moreover, obtained microphotographs were used to determine the mitochondria morphology. The analysis was performed with MicroP software (ver. 1.1.11b, Biomedical Image Informatics Lab, Taipei City, Taiwan (R.O.C.) Institute of Biomedical Informatics, National Yang Ming Chiao Tung University) powered by MATLAB (version R2010b, TheMathWorks, Natick, MA, USA) [27]. Based on confocal images of the mitochondrial network, the software automatically classified mitochondria morphology, categorizing them into proper subtypes with quantitive analysis. In order to measure mitochondrial dynamics and morphology, four microphotographs were used with two cells captured under 1000× magnification. 

In order to determine mitochondrial gene expression, mitochondria were isolated using the Mitochondria Isolation Kit for Cultured Cells (Thermo Fisher Scientific, Warsaw, Poland). All steps of the procedure were performed following the manufacturer’s instructions. The mitochondria were isolated from 2 × 10^7^ cells. The obtained mitochondria were homogenized with 1 mL of Extrazol^®^ (Blirt DNA, Gdansk, Poland). 

### 2.5. Evaluation of Eca EPCs Viability, Metabolism, and Oxidative Status

The cellular health of Eca EPCs non-OB and Eca EPCs OB was evaluated using Guava^®^ Muse^®^ Cell Analyser (Sigma-Aldrich/Merck, Poznan, Poland). The metabolic activity of cells was determined based on mitochondrial membrane potential using Muse^TM^ Mitopotential Assay Kit. Apoptosis profile was established in the test with Muse^®^ Annexin V and Dead Cell Kit. In addition, oxidative status was evaluated based on the intracellular accumulation of reactive oxygen species (Muse^®^ Oxidative Stress Kit) and nitric oxide (Muse^®^ Nitric Oxide Kit). All assays were performed according to protocols provided by the manufacturer (Luminex/Merck, Poznan, Poland) and were described in detail previously [28,29,30,31].

### 2.6. Determination of Transcripts Levels

The experimental cultures of Eca EPCs and isolated mitochondria were homogenized using Extrazol^®^ (Blirt DNA, Gdansk, Poland). Total RNA and mitochondrial RNA were isolated using the phenol-chloroform method. Their genomic DNA (gDNA) residuals were removed from samples with DNase I from PrecisionDNAse kit (Primerdesign, Blirt DNA, Gdansk, Poland). Purified RNA (500 ng) was transcribed into cDNA with Tetro cDNA Synthesis Kit (Bioline Reagents Limited, London, UK) to analyze genes expression. Both digestion of gDNA and reverse transcription (RT) was performed using T100 Thermal Cycler (Bio-Rad, Hercules, CA, USA). Quantitative PCR (qPCR) was performed using 1 μL of cDNA and specific primers (sequences listed in Appendix A). Two-tailed RT-qPCR was used for specific detection of miRNA levels. The exact conditions of reactions were described elsewhere [11], while primers are listed in Appendix A. All qPCR measurements were carried out using CFX Connect™ Real-Time PCR Detection System (Bio-Rad, Hercules, CA, USA). The obtained gene expression results were calculated using RQ_MAX_ algorithm and converted into the log2 scale as described previously [11,24,30].

### 2.7. Detection of Protein Expression with Western Blot

In order to determine the intracellular accumulation of selected proteins, cells were lysed using ice-cold RIPA buffer containing 1% of protease and phosphatase inhibitor cocktail (Sigma Aldrich/Merck, Poznan, Polska). The total concentration of proteins in samples was determined using the Bicinchoninic Acid Assay Kit (Sigma Aldrich/Merck, Poznan, Poland). The samples containing 20 μg of protein were mixed with 4× Laemmli loading buffer (Bio-Rad, Hercules, CA, USA). Subsequently, samples were incubated for 5 min at 95 °C in T100 Thermal Cycler (Bio-Rad, Hercules, CA, USA). Samples were separated in 12.5% sodium dodecyl sulphate-polyacrylamide gel electrophoresis (SDS-PAGE; 100 V for 90 min) in Mini-PROTEAN Tetra Vertical Electrophoresis Cell (Bio-Rad, Hercules, CA, USA) and transferred into polyvinylidene difluoride (PVDF) membrane in 1× Transfer buffer (Tris-base/Glycine/Methanol, Sigma-Aldrich/Merck, Poznan, Poland) using the Mini Trans-Blot^®^ system (Bio-Rad, Hercules, CA, USA) at 100 V for 60 min. The membranes were blocked using 5% bovine serum albumin (BSA) in TBS-T for 60 min. Detection of protein was performed by overnight incubation at 4 °C with primary antibodies. The incubation of membranes with secondary antibodies was performed for 60 min at 4 °C. After each incubation with the antibody, the membranes were washed five times for 5 min with TBST buffer.

Details of antibodies used for the reaction are presented in Appendix A. Chemiluminescent signal was detected using Bio-Rad ChemiDoc™ XRS system (Bio-Rad, Hercules, CA, USA) using DuoLuX^®^ Chemiluminescent and Fluorescent Peroxidase (HRP) Substrate (Vector Laboratories, Biokom, Janki, Poland). The Image Lab™ Software (Bio-Rad, Hercules, CA, USA) was used in order to analyze the molecular weight and intensity of signals. 

### 2.8. Immunocytochemical Detection

Immunocytochemical (ICC) staining was performed using previously described protocols [24,31]. Here, Eca EPCs cultures were performed in 24-well dishes covered by glass cover slides. Cultures were fixed with 4% PFA after 72 h of propagation. The incubation was performed 30 min at room temperature. Subsequently, specimens were washed three times using HBSS and prepared for permeabilization with 0.2% PBS-Tween solution supplemented with 10% goat serum. Specimens were permeabilized for 1 h. The incubation of samples with the primary antibody in HBSS was performed overnight at 4 °C. The secondary antibody was used at a concentration equal to 1:1000 in HBSS. Details of antibodies used for the reaction are presented in Appendix A. Each step of the protocol was preceded by gentle washing of specimens with HBSS. After staining, slides were fixed with a mounting medium with DAPI (ProLong™ Diamond Antifade Mountant with DAPI, Thermo Fisher Scientific, Warsaw, Poland). Observations were made using a confocal microscope (Leica TCS SPE, Leica Microsystems, KAWA.SKA Sp. z o.o., Zalesie Gorne, Poland). Images were processed and analyzed, as was previously described [24]. 

### 2.9. Statistical Analysis

The results obtained in the study are presented as the mean with standard deviation (±SD). The values are derived from at least three technical repetitions. The normality of the population data was determined using Shapiro–Wilk test, while equality of variances was assessed by Levene’s test. The measurements were conducted with STATISTICA 10.0 software (StatSoft, Inc., Statistica for Windows, Tulsa, OK, USA). Comparative analysis was performed using t-Student test or One-way analysis of variance with Dunnett’s post hoc test. These calculations were performed using GraphPad Software (Prism 8.20, San Diego, CA, USA). Differences with a probability of *p* < 0.05 were considered significant.

## 3. Results

### 3.1. Characterization of the Model Used for the Studies

The animals used for the experiment were classified using a system proposed by Henneke et al. [20]. Significant differences in terms of BSC between non-obese (non-OB) and obese (OB) groups were noted, warranting proper classification of animals (Appendix A). 

Furthermore, we have evaluated the phenotype of the cells used for the study and tested their multipotent character. The mesenchymal stem cells markers were assessed using RT-qPCR and with immunocytochemistry staining (Figure 1). The population of progenitor cells isolated from equine endometrium (Eca EPCs) consisted of cells with features characteristic for cells of mesenchymal origin, i.e., expression of markers CD44, CD90, and CD105 and low expression of hematopoietic markers, i.e., CD34 and CD45 (Figure 1b,c). Progenitor cells from the endometrium of non-obese mares and obese mares (Eca EPCs non-OB and Eca EPCs OB) differed in CD105 expression (Figure 1). The Eca EPCs OB showed decreased levels of mRNA for *CD105*. Moreover, the reduced transcript levels for *CD105* noted in Eca EPCs OB correlated with a decreased mRNA expression for *CD90* (Figure 1a,b). Both Eca EPCs non-OB and OB showed mRNA expression for perivascular markers *CD146* and *NG2* proteoglycan (neural/glial antigen 2), an epithelial marker, i.e., mucin-1 and smooth muscle markers, including actin alpha 1 (*ACTA2*), calponin 1 (*CNN1*) and myosin heavy chain 11 (*MHY11*). Notably, Eca EPCs OB were characterized by increased accumulation of transcripts for *ACTA2* (Figure 1a,b). 

Additionally, Eca EPCs used in the study underwent specific differentiation under osteogenic, chondrogenic, and adipogenic conditions, indicating their multipotency (Figure 2). However, the analysis of the extracellular matrix showed that Eca EPCs OB had lowered osteogenic and chondrogenic potential compared to Eca EPCs non-OB (Figure 2a–c). The adipogenic potential of Eca EPCs OB was also decreased (Figure 2d). 

### 3.2. Influence of Obesity on Proliferation of Equine Endometrial Progenitor Cells

The study showed that the progenitor cells isolated from the endometrium of obese mares (Eca EPCs OB) were characterized by decreased proliferative potential. In contrast, endometrial progenitor cells isolated from non-obese mares (Eca EPCs Non-OB) showed faster initial growth and enhanced proliferation capacity (Figure 3). The lowered self-renewal potential of Eca EPCs OB was demonstrated by a significantly reduced ability to form colony-forming units (CFUs) (Figure 3a,b). The migratory and wound healing assay revealed that Eca EPCs from OB were characterized by lowered expansion potency (Figure 3c–f), confirming lower CFU efficiency. Reduced proliferation of Eca EPCs OB in culture was reflected by the prolonged time needed for cell population doubling (PDT; Figure 3g). The metabolic activity of EPCs OB was also disturbed compared to Eca EPCs from non-obese mares. The significant difference between the metabolism of Eca EPCs from non-obese and obese mares was noted after 48 h of culture (Figure 3h). 

Cell cycle analysis confirmed that Eca EPCs OB had decreased potential for division (Figure 4). The intracellular accumulation of Ki67, a marker of actively proliferating cells, was significantly decreased in EPCs OB cultures (Figure 4a–c). Furthermore, the percentage of actively proliferating cells in Eca EPCs OB cultures, evidenced by a decreased S-phase ratio, shifted toward G0/G1-phase (Figure 4d,e). Reduced Ki67 protein expression and proliferation were correlated with the lower levels of microRNAs from the let7 family, known regulators of genes associated with proliferation of progenitor cells, i.e., let-7b and let-7c. However, let-7a expression was increased in Eca EPCs OB (Figure 4f–h).

### 3.3. The Influence of Obesity on Cytophysiology of Equine Progenitor Cells 

#### 3.3.1. Morphology, Growth Pattern, and Markers Associated with Cytoskeleton Assembly and Self-Renewal of Cells

Analysis of cell morphology and ultrastructure showed that Eca EPCs isolated from non-obese and obese mares showed fibroblast-like, spindle-shaped morphology (Figure 5a,b). Eca EPCs isolated from non-obese mares formed dense colonies in which well-developed intracellular connections were observed. In contrast, Eca EPCs OB formed less abundant aggregations, and intracellular connections between those cells were less marked. The actin cytoskeleton, as well as mitochondrial network, were more developed in EPCs from non-obese mares. The identification of senescence cells revealed that Eca EPCs from OB mares show significantly increased activity of β-galactosidase (Figure 5f). Additionally, mRNA levels for vimentin (*VIM*) and type III intermediate filament (IF) were decreased in Eca EPCs OB (Figure 5c). The lowered transcript levels of *VIM* noted for Eca EPCs OB, were correlated with decreased levels of miRNA-29a-3p and miR-181-5p, which function as regulators of cytoskeleton assembly and self-renewal of stem cells (Figure 5d,e).

#### 3.3.2. Mitochondrial Metabolism

Evaluation of mitochondria morphology and structure performed under the confocal microscope revealed the differences in the development of mitochondrial network between Eca EPCs derived from non-obese and obese mares (Figure 6). First of all, we noted an increased number of mitochondria measured per cell in endometrial progenitors derived from non-obese mares (Figure 6a–c). Analysis of mitochondria morphology showed that EPCs derived from non-obese mares show the increased presence of morphotypes associated with efficient energy production, including tubular-shaped, elongated, and branched mitochondria occurring during the fusion process (Figure 6d–g). Moreover, three-dimensional (3D) reconstruction revealed that a well-developed mitochondrial network characterizes the EPCs non-obese, while EPCs from obese mares showed a limited distribution of the mitochondrial network and were distinguished by the presence of respiratory deficient, globular mitochondria, which occurrence is linked with the accumulation of reactive oxygen species and lowered viability of cells (Figure 6a,b,h). 

Further, analysis of mitochondrial membrane potential (MMP) confirmed the lowered metabolism of Eca EPCs from obese mares (Figure 7). The significantly decreased mitochondrial metabolism was also correlated with the declining viability of cells. Gene expression patterns confirmed that the metabolism of mitochondria in EPCs OB was disturbed, with downregulation of transcripts essential for mtDNA function and the mitochondrial respiratory chain stability (Figure 8).

Analysis of parkin RBR E3 ubiquitin-protein ligase (PARKIN), PTEN-induced kinase 1 (PINK1), and mitofusin 1 (MFN1) expression was performed at mRNA and protein levels, both in Eca EPCs non-OB and OB (Figure 9). The mRNA level of *PARKIN* was decreased, while PARKIN protein was accumulated within Eca EPCs OB (Figure 9a–c). In contrast, *PINK* mRNA levels were elevated, but protein expression was lower in Eca EPCs OB (Figure 9a,d,e). However, mitofusin expression was decreased in EPCs OB, both at mRNA and protein levels (Figure 9a,f,g). Transcript levels determined for fission (*FIS*) were also reduced in Eca EPCs OB compared to Eca EPCs derived from non-obese mares (Figure 9h).

To summarize, data obtained on mitochondrial activity of endometrial progenitor cells showed that obesity significantly affects cellular metabolism. The EPCs OB were characterized by lowered mitochondrial membrane potential and dynamics associated with the occurrence of fragmented mitochondria. Deterioration in mitochondrial function noted in EPCs OB was also similar to decreased expression of genes related to (i) mitochondrial DNA control, including TFAM, (ii) assembly and stabilization of mitochondria complexes in the electron transport chain of mitochondria, e.g., NDUFA9, COX4I1, OXA1L; (iii) mitigating cellular stress, i.e., PIGBOS1; (iv) mitochondrial protein synthesis, e.g., PUSL1 and MRPL24; as well as (v) biogenesis MTERF4 and PPARGC1β. 

Furthermore, the expression of mitofusin facilitating mitochondrial fusion was significantly lowered in endometrial progenitors derived from obese mares. Additionally, endogenously accumulated PARKIN and PINK1, constituting an axis that regulates several cellular processes, including mitochondrial biogenesis, apoptosis, oxidative stress, and protein misfolding, showed opposite expression patterns. In endometrial progenitor derived from obese mares, we have noted an increased expression of PARKIN that may selectively accumulate on impaired mitochondria. In contrast, the expression PINK1, crucial for mitochondrial motility, was decreased.

#### 3.3.3. Viability, Oxidative Stress, and Intracellular Homeostasis

Apoptosis profile analysis supported the result indicating lower viability of Eca EPCs derived from obese than non-obese mares (Figure 10). The Eca EPCs OB showed an increased percentage of early apoptotic cells compared to Eca EPCs non-OB (Figure 10a,b,e). Although we observed increased early apoptosis in EPC cultures obtained from obese mares, the number of total apoptotic cells was not different for OB and non-OB (Figure 10a,b,d). EPCs OB showed a pro-apoptotic gene expression pattern, with an elevated *BAX/BCL-2* ratio and increased mRNA expression for caspase-9 (*CASP-9*), *P53*, and *P21* (Figure 11).

The decreased viability of EPCs OB was associated with oxidative stress and accumulation of nitric oxygen (NO) and reactive oxygen species (ROS). Dysregulated oxidative stress balance noted in EPCs OB was correlated with decreased levels of miR-20a-5p and miR-133b-3p (Figure 12). 

Expression pattern of FOXP3 (Figure 13a,b,e) and SIRT1 (Figure 13a,c,f,g) confirmed the disturbance of oxidative status in EPCs OB. The mRNA level of FOXP3 noted in EPCs was not affected by the physiological status of mares (Figure 13b). However, we observed a significantly decreased intracellular accumulation of FOXP3 protein in EPCs OB (Figure 13a,e). This phenotype may be related to lower regulatory potential and accumulation of oxidative stress markers of EPC OB. Moreover, lowered expression of SIRT1 in EPCs OB supports lower metabolic activity (Figure 13a,c,f,g). Elevated mRNA levels for osteopontin (OPN) and accumulated intracellular protein for EPCs OB confirm the molecular phenotype associated with obesity Figure 13a,d,h–j). Significantly increased expression of OPN was noted for glycosylated protein, with a molecular weight equal to 66 kDa, and fragments with molecular weight equal to 40 kDa (Figure 13a,h,i).

We noted an increased constitutive expression of AKT/PI3K in Eca EPCs derived from obese mares (Figure 14). The analysis of mRNA levels determined for *AKT* and *PI3K* indicated the increased accumulation of those transcripts in Eca EPCs OB (Figure 14d,g). However, a statistically significant difference was noted only in *AKT* mRNA expression (Figure 14d). In turn, intracellular detection of AKT and PI3K revealed a significant increase in those proteins in Eca EPCs from obese mares (Figure 14a–c,e,f,h,i).

## 4. Discussion

The endometrium has gained attention as a source of progenitor cells with high proliferative potential. The endometrial progenitor cells (EPCs) are described as a multipotent population, i.e., able to differentiate into adipogenic, osteogenic, chondrogenic, and even myogenic cell lineages or neurons [32]. The pro-regenerative features of EPCs, with their self-renewal properties, make them a treatment option for endometrial healing. Their potential therapeutic application has been strictly associated with treating endometrial diseases in women, such as Asherman’s syndrome, characterized by endometrial scarring, fibrosis, and dense adhesions with occlusion of the uterine cavity and endometriosis [32,33]. However, because of the high cellular plasticity and multipotent properties of EPCs, their therapeutic effectiveness in regenerative medicine could be much broader. For instance, endometrial stem cells were proposed as an alternative therapeutic tool for Parkinson’s disease [34]. Furthermore, in veterinary regenerative medicine, EPCs-based therapies could be a possible treatment for subfertility caused by endometrial dysfunction, which is a severe problem in the horse breeding industry [35]. Endometrial diseases occur commonly in mares without a specific etiology.

Obesity and insulin resistance can affect reproduction in mares, impacting the embryo and uterus, as well as the oocyte [2]. We have previously shown that the regenerative potential of multipotent stromal cells isolated from equine adipose tissue depends on the animal’s physiological status [8,9,11,36]. Disturbed cellular metabolism affects the pro-regenerative potential of progenitor cells, decreasing their utility in autologous transplantations. Obesity is one of the factors that significantly affect the quality of progenitor cells and their regenerative potential [37]. Studies on murine model and human subjects, showed that obesity reduces self-renewal potential of adipose-derived stromal cells related to decreased telomerase activity [38]. Furthermore, mature adipocytes from obese individuals are characterized by elevated levels of mitogen-activated protein 4 kinase 4 (MAP4K4), which results in their decreased cellular plasticity [39].

In this study, we have shown that obesity affects the molecular phenotype of equine endometrial progenitor cells (Eca EPCs) and may reduce their multipotency. We observe, inter alia, that Eca EPCs from obese mares had significantly decreased mRNA expression for CD105 surface marker (endoglin) compared to Eca EPCs from non-obese mares. The CD105 is an essential factor associated with the differentiation potential of adult stem cells originating from mesenchymal tissues, i.e., multipotent stromal/stem cells (MSCs) [40]. The expression of this marker varies between MSCs of different origins and is related to their heterogeneity [41]. Indeed, the population of Eca EPCs OB used in this study had lowered capacity for osteogenic and chondrogenic differentiation. The functional role of CD105 in MSC biology is still deliberated, especially its effect on chondrogenesis in vitro [42]. Thus, future studies related to multipotency of Eca EPCs should also include an analysis of CD105 association with their plasticity and potential for tissue-specific differentiation. Notably, the increased mRNA expression for α-smooth muscle actin (ACTA/α-SMA) was also a characteristic of Eca EPCs OB. Studies by Talele et al. show that α-SMA is a feature of human mesenchymal stromal cells with lowered clonogenic potential and limited cellular plasticity, confirming the results obtained in this study [43].

Essentially, the current study demonstrated that obesity reduces the proliferation and mitochondrial metabolism of equine endometrial progenitor cells (Eca EPCs). The regenerative potential of progenitor cells is reflected by high proliferative (self-renewal) capacity and viability [37]. We have demonstrated that Eca EPCs isolated from mares that are obese have decreased clonogenic and proliferative potential, as well as migratory activity. Lower mitochondrial metabolism in Eca EPCs was associated with apoptosis and oxidative stress. Our results correspond with the study of Alessio et al. [44], who showed that obesity in mice not only affects the proliferation of progenitor cells isolated from subcutaneous and visceral adipose tissue but also affects cells derived from bone marrow.

In our study, Eca EPCs from obese mares had a reduced percentage of progenitor cells in the S-phase of the cell cycle, associated with a lower potential for cell division and a prolonged time for population doubling. The clonogenic potential of EPCs from obese mares was also decreased compared to EPCs from non-obese mares, which agrees with the finding that obesity negatively impacts the clonogenicity of endometrial mesenchymal stem cells of women with reproductive failure [45]. In previous studies, we observed that the proliferative potential of multipotent adipose-derived stromal cells (ASC) could be affected by increased age and the occurrence of equine metabolic syndrome. An increased percentage of G1/G0-arrested cells was characteristic for cells isolated from middle-aged horses (5–15), old (>15 years old), or had metabolic syndrome [46]. Indeed, the progenitor cells of mesenchymal origin isolated from visceral adipose tissue and bone marrow of obese mice were characterized by decreased proliferation rate linked to a reduction of S-phase cells together wi increased senescence occurrence.

The decreased self-renewal potential of EPCs isolated from obese mares was correlated with cell-cycle arrest and reduced expression of Ki67, a marker of actively proliferating cells. Obesity increases Ki67 expression in human progenitor cells from adipose tissue and smaller adipocytes as well as leads to their cellular death [47]. A hallmark of the inflammation accompanying obesity is a reduction in Ki67 expression and inhibited proliferation of progenitor cells in the taste buds [48]. Thus, obesity-induced changes in Ki67 levels can depend on tissue niche and cell type. For instance, functional endometrium is characterized by the presence of stromal cells showing increased Ki67 expression in addition to increased levels of the anti-apoptotic protein, BCL-2 [49].

We obtained a similar expression profile in our study, as EPCs isolated from obese mares exhibited a phenotype of early apoptotic cells, with increased mRNA levels for pro-apoptotic BAX, caspase 9, P53, and P21 and decreased expression of BCL-2. The balance between apoptosis and proliferation in endometrial cells is essential to regulate the regeneration of the endometrium during the reproductive cycle. It is worth mentioning that accelerated proliferation ratio with simultaneous apoptosis resistance also characterizes endometrial stromal cells during pathological processes, including adenomyosis development. Thus the knowledge related to the regulation of the endometrial progenitor cell viability and proliferative status is essential for endometrium homeostasis [50].

Decreased viability of Eca EPCs OB noted in this study can be related to the lowered expression of mRNA for vimentin (*VIM*). Generally, vimentin is recognized as a type III intermediate filament protein abundant in tissues of mesenchymal origin. This cytoskeleton protein plays a pivotal role in maintaining the shape and integrity of cells. It is also notable that vimentin is crucial for tissue homeostasis, protecting cells from protein misfolding stress [51]. Given the above, the Eca EPCs OB show features of early apoptotic cells with decreased cellular function, including self-renewal potential.

Our study showed, for the first time, that progenitor cells isolated from the endometrium of obese mares during anestrus have a pro-apoptotic profile and lower mitochondrial potential than cells from non-obese mares. The molecular phenotype of obese endometrium was also consistent with an increased expression of osteopontin (OPN). During obesity, high expression of OPN could correlate with the development of insulin resistance. OPN is depicted as a biomarker for obesity-associated inflammation and deteriorating adipose tissue metabolism related to enhanced macrophage infiltration, which was noted using the mice model [52]. Knockout of OPN in mice endometrial stromal cells resulted in lower adhesion and invasion of the blastocyst, as well as decreased expression of matrix metalloproteinase-9 (MMP-9) associated with impaired invasive competency of the trophoblast [53]. Increased levels of OPN are also correlated with oxidative stress and decreased mitochondrial membrane potential, which was observed using the model of mouse cardiac myocytes [54]. Collectively, our results showed that OPN might be associated with the deterioration of Eca EPCs metabolism induced by obesity. Thus, OPN may serve as a potential target in terms of developing novel therapeutic strategies for endometrial diseases.

Furthermore, analysis of mitochondrial dynamics revealed the increased occurrence of fragmented globular mitochondria in EPCs derived from the obese endometrium compared to EPCs from non-obese mares characterized by elongated tubular mitochondria. More recently, the association between mitochondria morphology, cellular status, and processes has been studied extensively. The changes in the morphology of mitochondria are dynamic, allowing for optimization and precisely tuning mitochondria function to support cellular demands [55]. Accumulating evidence has revealed that increased membrane potential and elongated mitochondria, forming filamentous structures associated with variable degrees of branching or reticulation, reflect the healthy respiratory status of the cells and serve increased cellular energy production [56,57,58]. In this manner, the elongated mitochondria act as intracellular power-transmitting cables [59]. 

At the same time, changes in mitochondria physiology induced by oxidative stress are associated with loss of mitochondrial membrane potential and result in small and globular mitochondria formation [60]. In addition, Wang et al. showed that elongated mitochondria are more resistant to deterioration induced by reactive oxygen species (ROS) and mitophagy than fragmented mitochondria [61]. It was also demonstrated that elongated mitochondria could cope with increased ROS production as they have elevated oxidative phosphorylation activity and thus developed a mechanism that allows them to resist an acute burst of superoxide [61,62].

Obesity-induced morphological and functional changes of mitochondria have been noted in skeletal muscle and linked with impaired tissue metabolism [63]. Metabolic imbalance related to decreased mitochondrial activity and oxidative stress is linked with disturbed decidualization, the pathogenesis of endometriosis, and endometrial cancer development [64].

The cellular stress noted in endometrial progenitor cells is related to the pro-inflammatory uterine environment. In the current study, expression of FOXP3 and SIRT1 was lower in progenitor cells isolated from obese than non-obese endometrium. A characteristic of regulatory T cells, FOXP3 is responsible for the immunosuppressive capacity of multipotent stromal cells and may participate in regulating immune responses in endometriosis [65]. Endometrial FOXP3 in women with endometriosis decreases linearly from the early to late proliferative phase [66]. Additionally, using the Foxp3GFP mice model, Tales et al. showed that CD4^+^CD25^+^ Foxp3+ regulatory T cells (Tregs) are needed for successful implantation of the blastocyst in the uterus because they prevent the development of a hostile uterine microenvironment [67].

In turn, SIRT1 suppresses nuclear factor-κB (NF-κB), a transcription factor regulating the expression of various pro-inflammatory cytokines and chemokines [68,69]. Moreover, using a model of human endometrial cells, it was found that SIRT1 is an essential molecule for regulating embryo implantation into the endometrium through stimulation of E-cadherin [70].

Compared to normal-weight mares, embryos from obese mares have higher expression of transcripts associated with inflammation, endoplasmic reticulum, oxidative and mitochondrial stress [6]. As shown in the nonhuman primate model, defects of mitochondrial metabolism occur in endometriosis with a decrease in electron transport chain complex I- and II-mediated mitochondrial respiration [71], which correlates with our study. Comparative analysis showed that EPCs OB compared to EPCs non-OB had decreased expression of transcripts essential for mtDNA function and mitochondrial respiratory chain stability.

Defects in mitochondrial metabolism observed in EPCs OB could originate from the accumulation of reactive oxygen species (ROS) and nitric oxide, which is also accompanied by decreased mRNA and protein expression of PTEN-induced kinase 1 (PINK1) and mitofusin 1 (MNF1). PINK1 is an essential regulator of mitochondrial homeostasis and energy metabolism. Studies performed using the PINK1-knockout mice model showed that mammalian PINK1 is a key factor protecting against oxidative stress in the mitochondrion [72]. Decreased levels of PINK1 are noted in the endometrium with minimal and mild endometriosis [73]. Mitofusins, including mitofusin 1 and 2 (MFN1 and MFN2, respectively), are crucial mediators of mitochondrial fusion, which compensate for the functional defects of the mitochondrion. Moreover, MFN1 is particularly important in terms of female fertility. The absence of MFN1 is associated with oocyte apoptosis and cell-cycle arrest, depletion of the ovarian follicular reserve, and characteristics of female reproductive aging [74]. Moreover, increased expression of PARKIN noted in Eca EPCs OB can be associated with clearance of dysfunctional mitochondria and activation of mechanisms that protect cells against apoptosis [75].

We also noted that obesity-induced changes in endometrial progenitor cells were associated with increased constitutive expression of PI3K/AKT and increased β-galactosidase activity. The PI3K/AKT pathway controls various critical cellular processes, including glucose homeostasis, lipid metabolism, and protein synthesis; it also affects cell proliferative activity and viability. Increased AKT expression is related to increased adipogenesis and the obese phenotype [76]. An overactive AKT pathway is a characteristic of endometrial cancer and endometriosis; thus, regulation of PI3K/AKT signaling also became a consideration in the matter of therapeutic strategies for endometrial disorders [77].

Obesity induces cellular senescence in mesenchymal stromal/stem cells, related to a chronic inflammatory condition [7]. In a model of high-fat diet (HFD) fed obese mice, Gao et al. showed that this chronic inflammatory status is linked to sustained mitogen-activated protein kinase (MAPK) signaling and nuclear factor kappa B (NF-κB) activation, resulting in pro-inflammatory cytokine upregulation [78].

Consistent with our results, Conley et al. showed that obesity elicits an early senescence program in progenitor cells, manifested by an increased expression of such markers as p53 and p21. However, in the study, Coloney et al. indicated that senescence-associated beta-galactosidase activity (SA-β-gal) is not significantly increased in adipose-derived MSCs of obese subjects. In contrast, we found that EPCs from obese mare endometrium accumulated significantly more SA-β-gal in comparison to EPCs isolated from non-obese endometrium.

Using two-tailed RT-qPCR [79,80], we established equine non-coding RNA levels. We have tested miRNAs regulating oxidative status and counteracting the senescent cell burden of progenitor cells. The obtained miRNA profile confirmed that obesity attenuates the molecular phenotype of EPCs, resulting in decreased proliferative activity and oxidative stress. Eca EPCs from obese endometrium were characterized by decreased levels of let-7b and let-7c, as well as miR-20a-5p, -29a-3p, -133b-3p and -181a-5p, but significantly increased levels of let-7a. Loss of let-7b expression is connected with endometriosis pathophysiology and inflammation [81]. In human adipose-derived stem cells, let-7c is recognized as a molecule promoting ectopic bone formation and suppressing adipogenesis via high-mobility group AT-hook 2 [82]. Thus, decreased expression of let-7b and let-7c in EPCs from obese endometrium could be related to the deterioration of self-renewal potential and decreased cellular plasticity. Moreover, increased levels of let-7a may reflect the lowered immunomodulatory potential of Eca EPCs OB, as linked to decreased expression of FOXP3 [83]. Lowered proliferative potential of Eca EPCs OB was also confirmed by decreased levels of miR-29a and miR-181a-5p. Those molecules are not only essential regulators of the self-renewal potential of MSCs, but they also promote the wound healing capacity of tissues [84]. Furthermore, miR-29a promotes angiogenesis and protects against fibrosis post-injury, regulating the biological activity of MMPs and vascular endothelial growth factors (VEGFs) [85]. Down-regulation of miR-20a-5p and miR-133b-3p are observed in mesenchymal stem cells as a result of oxidative stress damage [86,87]. Moreover, miR-20a-5 was recognized as a suppressor of endometrial cancer progression by targeting janus kinase 1 (Jak-1), belonging to a class of protein-tyrosine kinases involved in malignancies [88]. Given the above, analysis of miRNA levels, both as tissue-specific and circulating markers, may bring additional insight into the mechanism of endometrial receptivity, regeneration, and pathology, thus gaining more and more attention from researchers [89].

While not the focus of this manuscript, our results marked differences in the endometrium of mares associated with obesity. These changes suggest that the obese mare’s uterus could be less able to adapt to cyclic changes and provide an optimal environment during pregnancy. While further studies are needed in this area, body condition is a potential consideration in broodmare management.

## 5. Conclusions

In summary, we demonstrated that obesity in the mare alters functional features of endometrial progenitor cells, such as molecular phenotype associated with their potential for specific differentiation, self-renewal activity, and mitochondrial metabolism and dynamics. Obesity attenuated the cellular homeostasis of endometrial progenitor cells, resulting in increased oxidative stress and hampered pro-regenerative signaling. Early apoptosis and increased senescence characterized progenitor cells when derived from obese endometrium. The impaired cytophysiology of progenitor cells from obese endometrium predicts lower regenerative capacity if used as autologous transplants. In our opinion, further studies are required to evaluate the cellular plasticity and immunomodulatory properties of endometrial progenitor cells. Future studies should also focus on developing methods to change the trajectory of obesity-induced metabolic deterioration to improve their functional properties.

## Figures and Tables

**Figure 1 cells-11-01437-f001:**
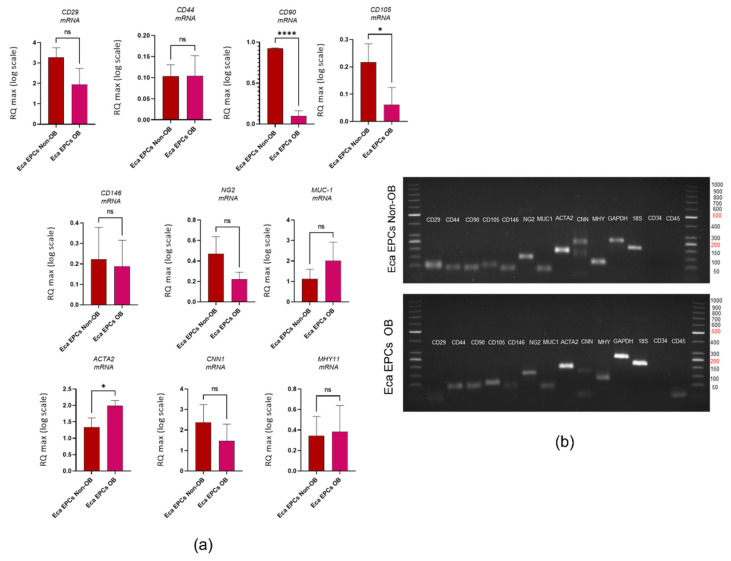
Phenotype of Eca EPCs. The RT-qPCR analysis aimed to determine relative transcript levels of cell surface markers (**a**). The mRNA expression was established for MSCs markers (CD29, CD44, CD90, CD105), perivascular markers (NG2, CD146), epithelial marker (MUC-1) and smooth muscle markers (ACTA2, CNN1, MHY11). The mRNA expression for hematopoietic markers (CD34, CD45) was not included on the graphs due to the lack of a specific signal. The PCR products were analyzed using electrophoresis (**b**). The cellular phenotype was tested using immunocytochemistry. The scale bar indicated on merged figures is equal to 30 μm (**c**). All results are shown as mean ± SD. Columns with bars represent means ± SD. * *p*-value < 0.05, **** *p*-value < 0.0001, while *ns* symbol refers to non-significant differences.

**Figure 2 cells-11-01437-f002:**
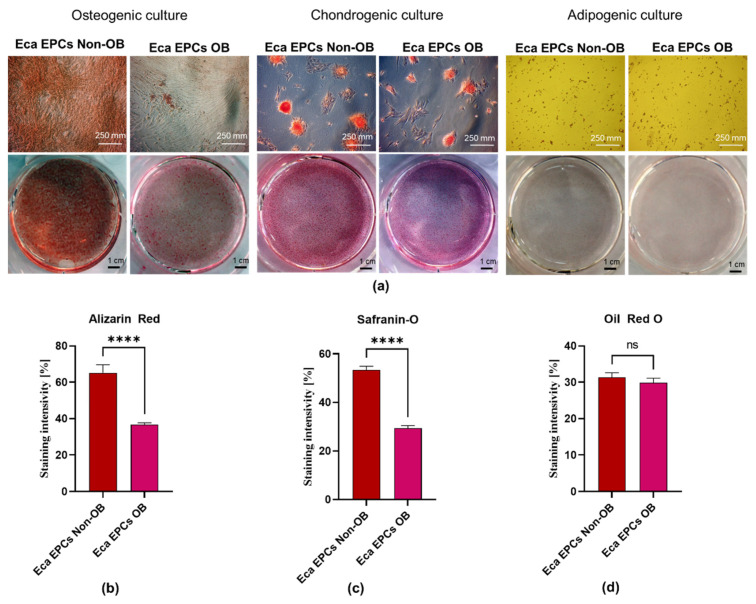
The analysis of Eca EPCs multipotency. Tissue-specific differentiation extracellular matrix features were documented (**a**). Cultures were maintained under osteogenic, chondrogenic and adipogenic conditions. Calcium deposits formed under osteogenic conditions were detected using Alizarin Red staining, while chondrogenic nodules were stained with Safranin-O dye. The lipid-rich vacuoles after adipogenic stimulation were detected with Oil Red O. The staining efficiency was measured to compare the differentiation potential of Eca EPCs non-OB with Eca EPCs OB (**b**–**d**). Columns with bars represent means ± SD. **** *p*-value < 0.0001, while *ns* symbol refers to non-significant differences.

**Figure 3 cells-11-01437-f003:**
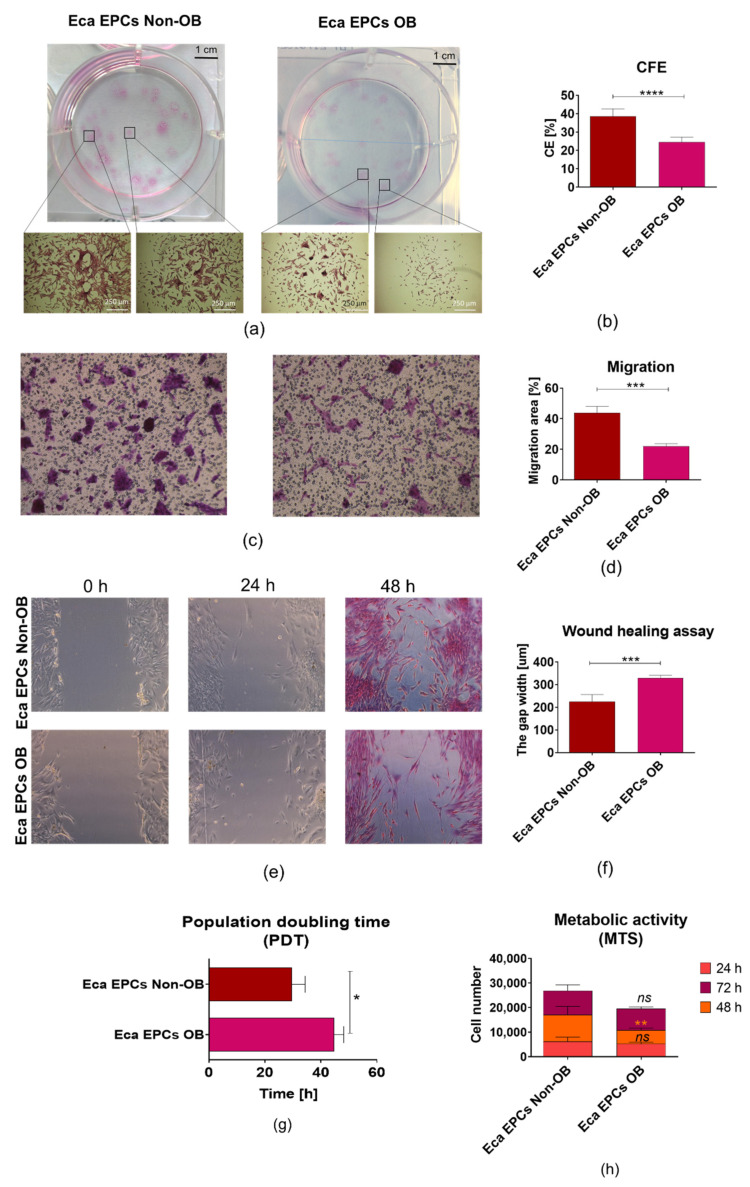
The proliferative capacity of equine endometrial progenitor cells (Eca EPCs) isolated from non-obese mares (non-OB) and obese mares (OB), expressed by colony formation capability (**a**,**b**), migratory efficiency (**c**–**f**), population doubling time (**g**), metabolic activity (**h**). Columns with bars represent mean ± SD. * *p*-value < 0.05, ** *p*-value < 0.01, *** *p*-value < 0.001 and **** *p*-value < 0.0001, while *ns* symbol refers to non-significant differences.

**Figure 4 cells-11-01437-f004:**
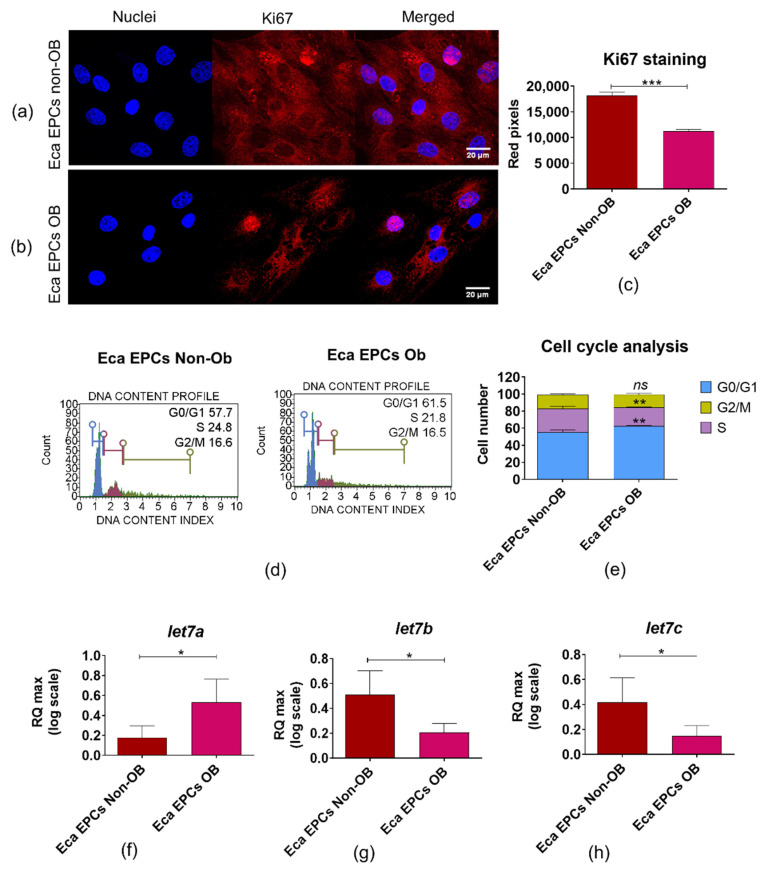
The proliferative capacity of equine endometrial progenitor cells (Eca EPCs) isolated from non-obese mares (non-OB) and obese mares (OB) evaluated based on Ki67 expression (**a**–**c**), distribution of cells within the cell cycle (**d**–**e**), and levels of particular let7 family members: *let7a* (**f**), *let7b* (**g**) and *let7c* (**h**). Columns with bars represent means ± SD. * *p*-value < 0.05, ** *p*-value < 0.01 and *** *p*-value < 0.001, while *ns* symbol refers to non-significant differences.

**Figure 5 cells-11-01437-f005:**
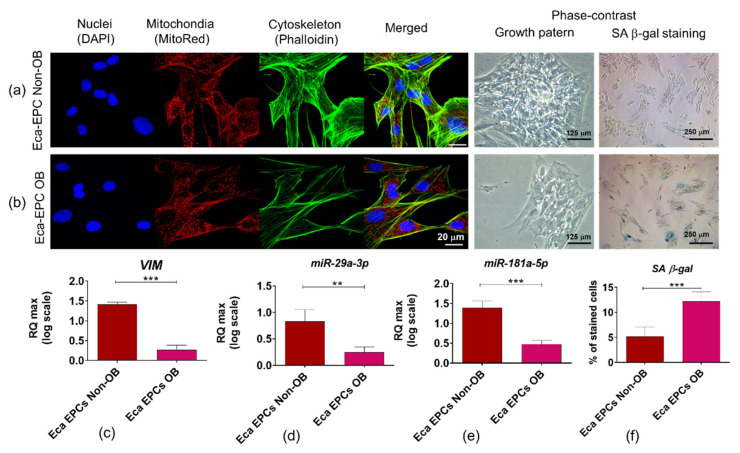
Morphology, ultrastructure, the growth pattern of equine endometrial progenitor cells (Eca EPCs) isolated from non-obese mares (non-OB) and obese mares (OB), and analysis of transcripts associated with a cytoskeletal network. The cultures were imaged with a confocal microscope to determine ultrastructure. Growth pattern of cell cultures was monitored under a phase-contrast microscope. Scale bars are indicated in the representative photographs (**a**,**b**). Additionally, RT-qPCR was performed to determine the mRNA expression of vimentin (*VIM*); (**c**) and levels of miR-29a-3p and miR-181-5p (**d**,**e**). Moreover, the senescence-associated β-Galactosidase (SA β-gal) was detected in cultures (**f**). Results are presented as a column with bars representing means ± SD. ** *p*-value < 0.01 and *** *p*-value < 0.001.

**Figure 6 cells-11-01437-f006:**
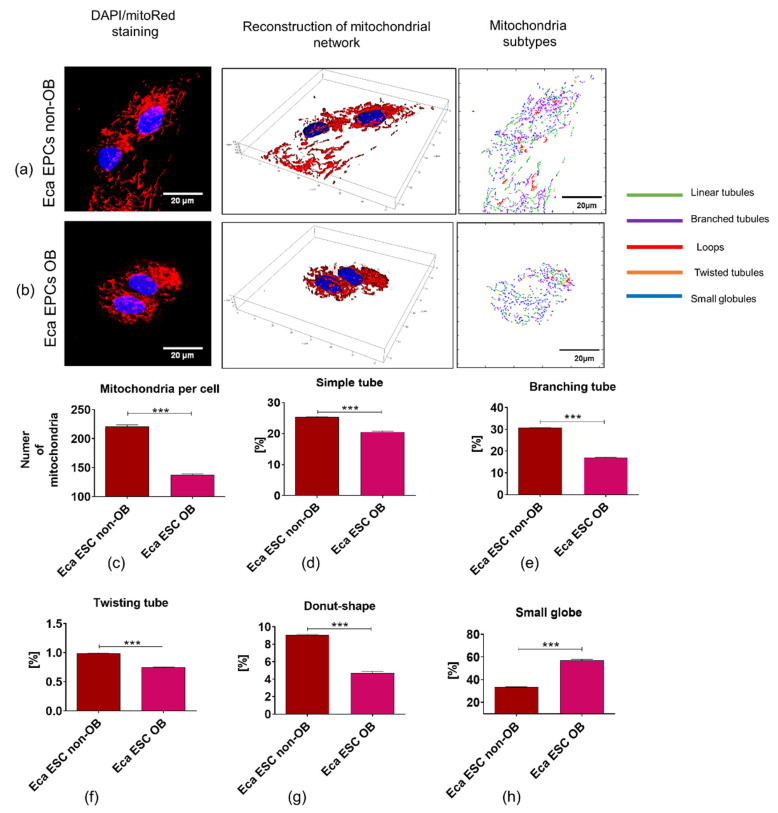
Mitochondrial network, numbers, and morphology of equine endometrial progenitor cells (Eca EPCs) isolated from non-obese mares (non-OB) and obese mares (OB). Imaging of Eca EPCs with high magnification allowed assessment of the mitochondrial net and its dynamics (**a**,**b**), the number of mitochondria per cell (**c**) and classification of mitochondrial morphology (**d**–**h**). Mean ± SD is presented as columns and bars. *** *p*-value < 0.001.

**Figure 7 cells-11-01437-f007:**
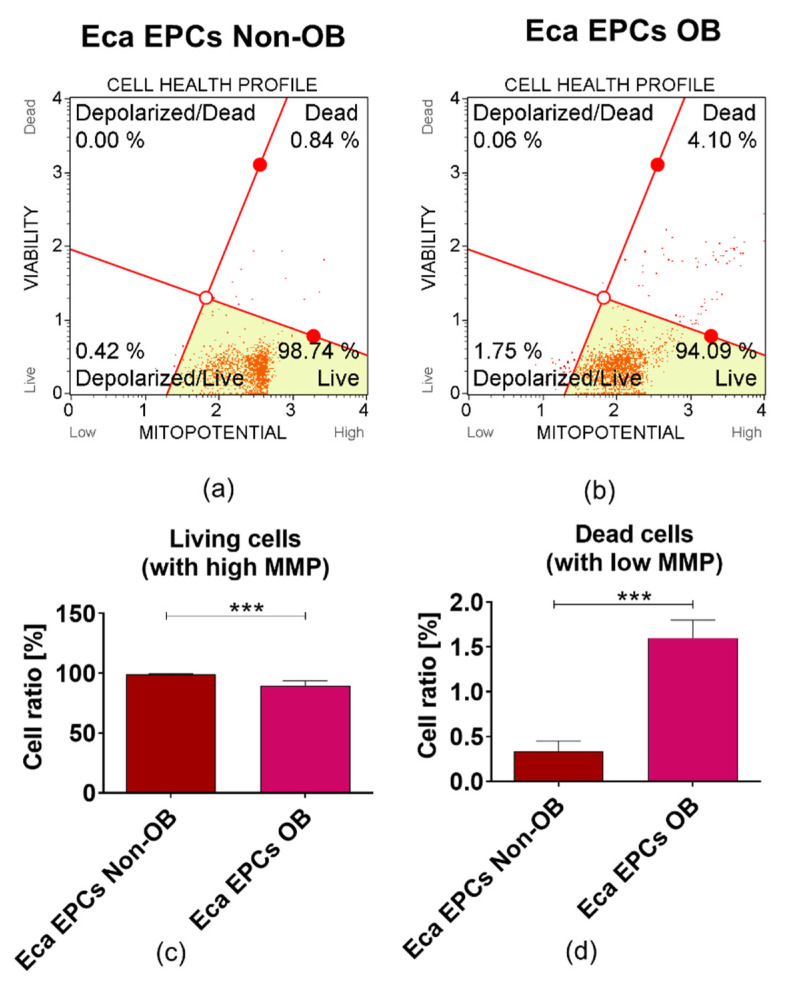
The mitochondrial membrane potential (MMP) determined in equine endometrial progenitor cells (Eca EPCs) isolated from non-obese mares (non-OB) and obese mares (OB) The representative graphs show the distribution of cells, taking into account their viability and mitochondrial membrane depolarisation (**a**,**b**). The cells were counterstained with cationic and lipophilic dye to detect changes in MMP and 7-AAD. The gating strategy allowed us to evaluate contribution of four populations of cells, i.e., (i) healthy cells with high MMP (live—bottom right corner); (ii) live cells with low MMP (depolarized/live—bottom left corner), (iii) dead (late apoptotic cells) with depolarized mitochondrial membrane (depolarized/dead—upper left corner) and necrotic (dead—upper right corner). Comparative analysis was performed to determine differences between EPCs non-OB and OB in terms of viability (**c**) and mitochondrial activity (**d**). Mean ± SD. *** *p*-value < 0.001.

**Figure 8 cells-11-01437-f008:**
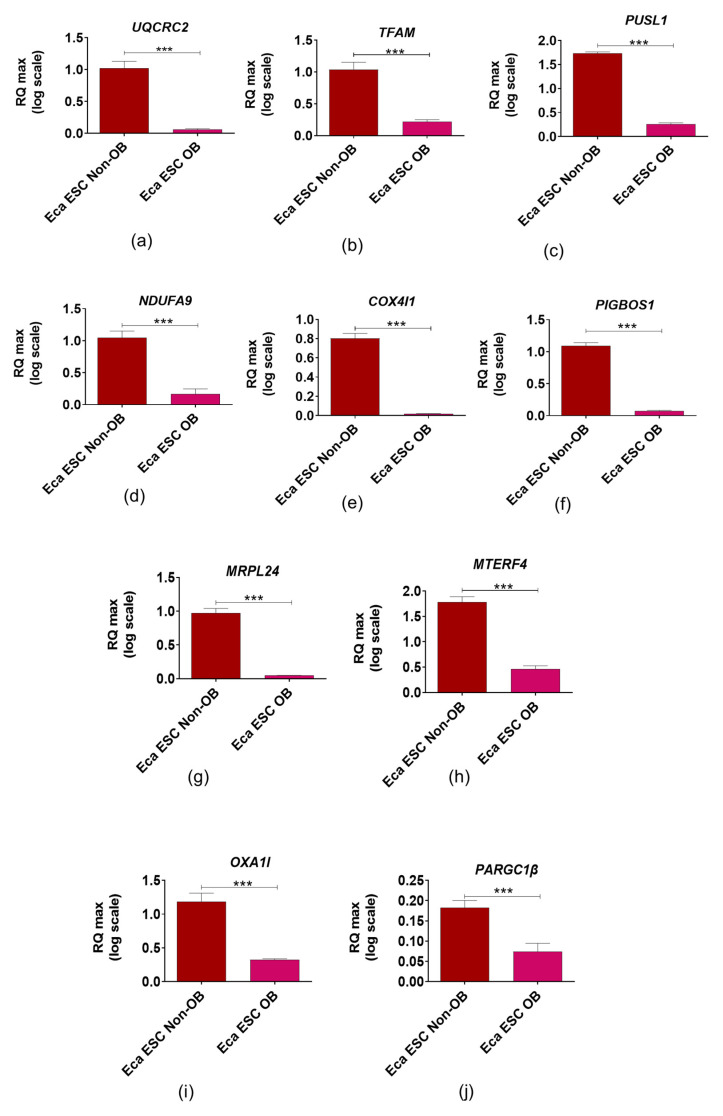
Transcript levels for genes associated with mitochondria homeostasis determined for equine endometrial progenitor cells (Eca EPCs) isolated from non-obese mares (non-OB) and obese mares (OB). The analysis was performed usi, bng RT-qPCR technologies. The following genes were measured: ubiquinol-cytochrome c reductase core protein 2 (UQCRC2, (**a**)), transcription factor A, mitochondrial (TFAM, (**b**)), pseudouridylate synthase-like 1 (PUSL1, (**c**)), NADH: ubiquinone oxidoreductase subunit A9 (NDUFA9, (**d**)), cytochrome c oxidase subunit 4I1 (COX4I1, (**e**)), PIGB opposite strand 1 (PIGBOS1, (**f**)), mitochondrial ribosomal protein L24 (MRPL24, (**g**)), mitochondrial transcription termination factor 4 (MTERF4, (**h**)), mitochondrial inner membrane protein (OXA1L, (**i**)) and PPARG coactivator 1 beta (PPARGC1β, (**j**)). Obtained data were normalized to the expression of the reference gene and expressed as using RQ (max) algorithm. Results are presented as columns with bars representing means ± SD. *** *p*-value < 0.001.

**Figure 9 cells-11-01437-f009:**
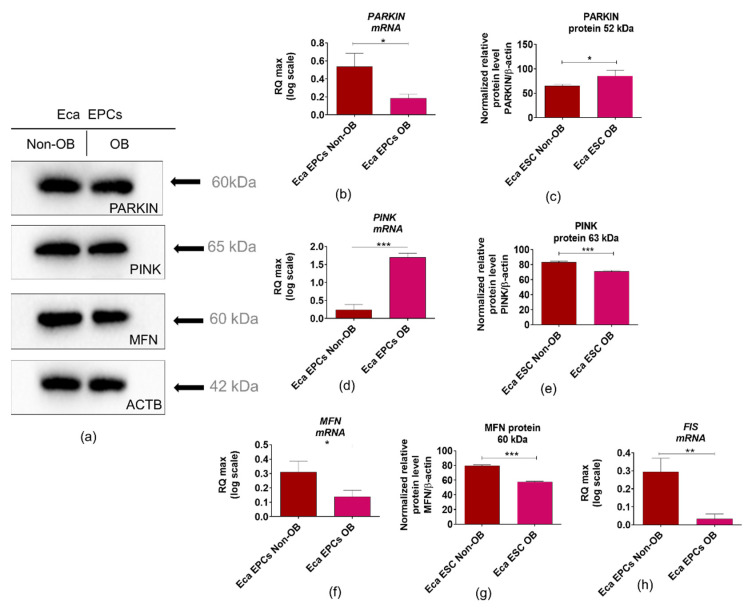
The expression profile of molecular markers associated with mitochondrial dynamics determined for equine endometrial progenitor cells (Eca EPCs) isolated from non-obese mares (non-OB) and obese mares (OB). Parkin RBR E3 ubiquitin-protein ligase (PARKIN), PTEN-induced kinase 1 (PINK1) and mitofusin 1 (MFN1) were determined based on mRNA (**b**,**d**,**f**) and protein levels (**a**)—representative blots; (**c**,**e**,**g**)—graphs reflecting the expression of proteins normalized to the β-actin/ACTB), additionally transcript levels for fission (FIS) were determined (**h**). Means ± SD. * *p*-value < 0.05, ** *p*-value < 0.01 and *** *p*-value < 0.001.

**Figure 10 cells-11-01437-f010:**
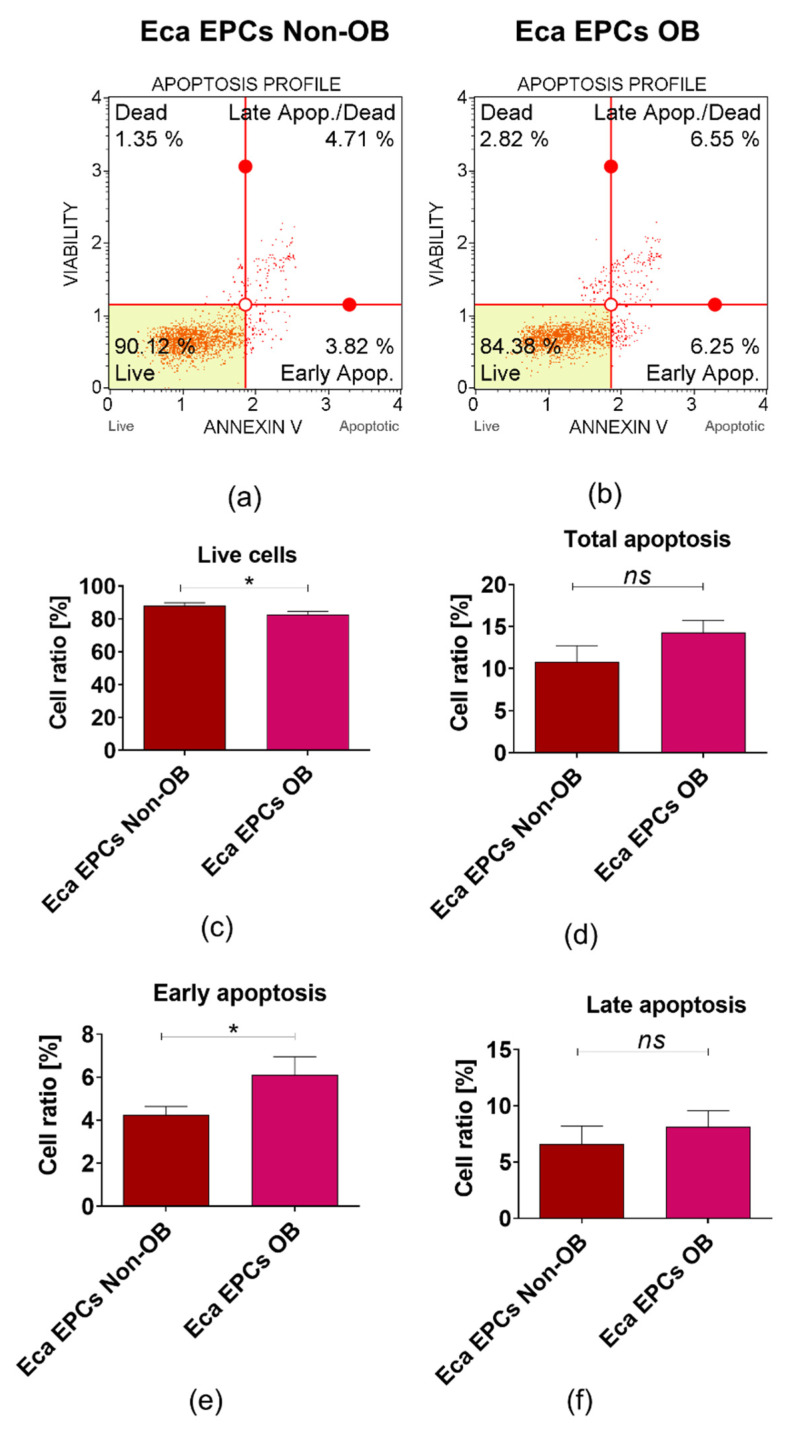
The apoptosis profile determined in equine endometrial progenitor cells (Eca EPCs) isolated from non-obese mares (non-OB) and obese mares (OB). The representative graphs show the distribution of cells, taking into account their viability (**a**,**b**). The viable cells are noted in the left bottom corner (Live), early apoptotic cells are located in the right bottom corner. Late apoptotic cells are visible in the right upper corner, while necrotic cells (dead) are in the left upper corner. The total apoptotic cells percentage reflects the sum of early and late apoptotic cells. Comparative analysis was performed to determine differences between Eca EPCs non-OB and OB for viability (**c**) total cell apoptosis (**d**) early apoptosis (**e**) and late apoptosis (**f**). Means ± SD. * *p*-value < 0.05, while ns symbol refers to non-significant differences.

**Figure 11 cells-11-01437-f011:**
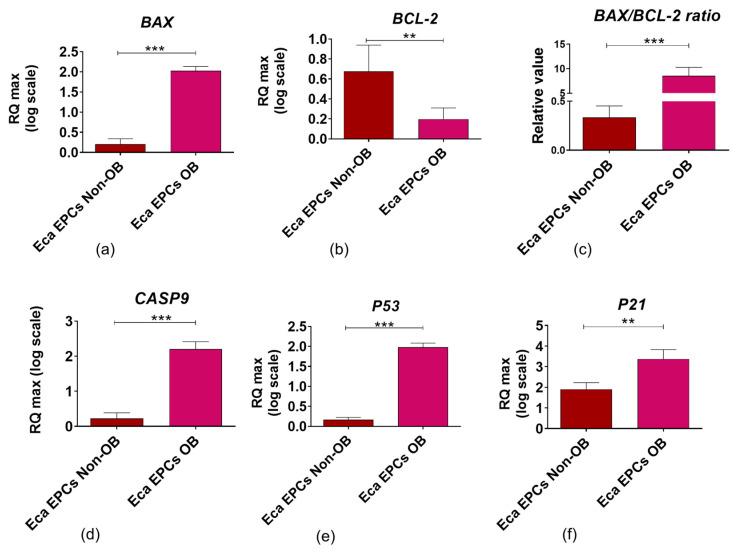
The expression profile of genes associated with apoptosis determined in equine endometrial progenitor cells (Eca EPCs) isolated from non-obese mares (non-OB) and obese mares (OB). The transcript levels were determined using RT-qPCR and normalized to the expression of *GAPDH* (glyceraldehyde-3-phosphate dehydrogenase) and spike control. The tested apoptotic markers were Bcl-2-associated X protein (*BAX*, (**a**)) and B-cell lymphoma 2 (*BCL-2*, (**b**)) which levels were used for determination of BAX/BCL-2 ratio (**c**). Moreover, mRNA levels were established for caspase 9 (*CASP9*, (**d**)), cellular tumor antigen p53 (*P53*, (**e**)) and cyclin-dependent kinase inhibitor 1A (*P21*, (**f**)). The relative values of gene expression were established using RQ_MAX_ algorithm. Means ± SD, ** *p*-value < 0.01 and *** *p*-value < 0.001.

**Figure 12 cells-11-01437-f012:**
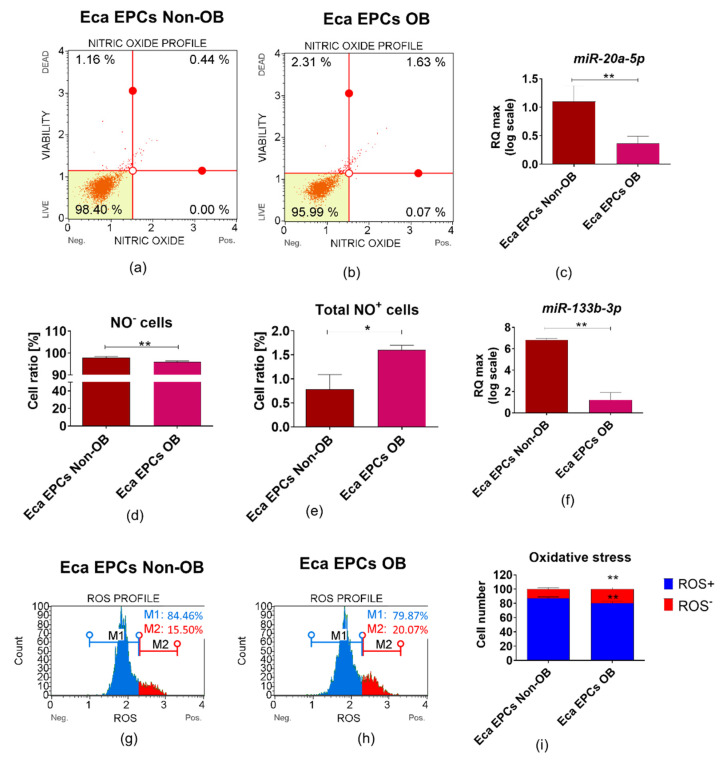
The analysis of the oxidative status in equine endometrial progenitor cells (Eca EPCs) isolated from non-obese mares (non-OB) and obese mares (OB). Representative dot plots show the distribution of cells based on intracellular accumulation of nitric oxide—NO (**a**,**b**), while column graphs show statistical analyses (**d**,**e**). The cells located in the left bottom corner of dot-plot graphs did not accumulate NO (Live), while viable cells that accumulated NO are located in the right bottom corner. Dead cells are located in the upper part of dot-plot—in the left corner NO negative cells (NO^−^) while in the right NO positive cells (NO^+^). The increased overall percentage of NO positive cells noted in EPCs OB corresponds with the lowered expression of miR-20a (**c**) and miR-133b (**f**). The distribution of cells based on reactive oxygen species (ROS) accumulation is visible on histograms (**g**,**h**). Cells that did not accumulate intracellular ROS are marked with a blue gate, while ROS positive cells are marked with a red gate. The comparative analysis confirmed oxidative stress in Eca EPCs OB (**i**). Means ± SD. * *p*-value < 0.05 and ** *p*-value < 0.01.

**Figure 13 cells-11-01437-f013:**
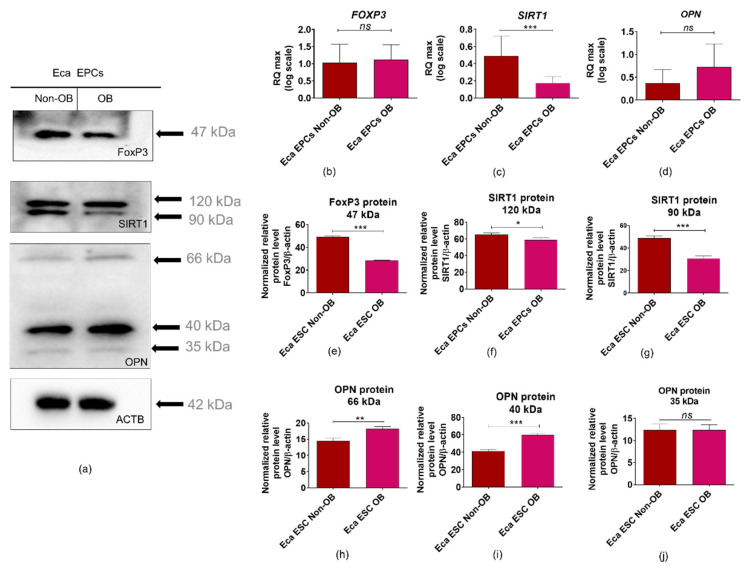
Expression of forkhead box P3 (FOXP3), sirtuin 1 (SIRT1) and osteopontin (OPN) determined in equine endometrial progenitor cells (Eca EPCs) isolated from non-obese mares (non-OB) and obese mares (OB). The protein levels were normalized to β-actin (ACTB). Protein expression was detected using Western blot technique (**a**), while RT-qPCR (**b**–**d**) was used to establish the mRNA level for genes of interest. The densitometry measurements of membranes were performed to compare the expression of proteins between EPCs non-OB and OB (**e**–**j**). Significant differences were calculated for normalized values and shown as means ± SD. * *p*-value < 0.05 and ** *p*-value < 0.01 and *** *p*-value < 0.001.

**Figure 14 cells-11-01437-f014:**
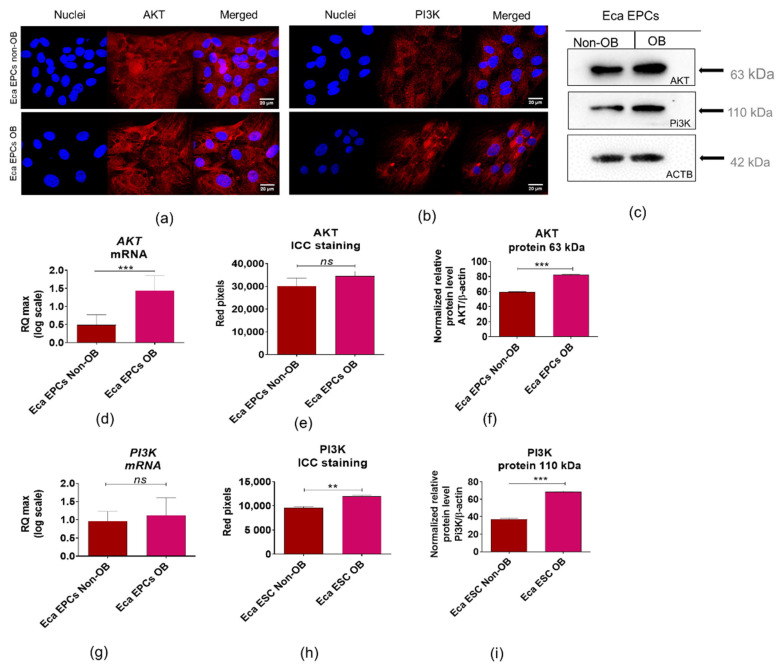
Expression of Pi3K/AKT determined and compared between EPCs non-OB and OB. The intracellular accumulation of AKT and PI3K was visualized using immunocytochemistry (ICC, **a**,**b**) and determined with Western blot (**c**). RT-qPCR (**d**–**i**) was used to establish mRNA levels for genes of interest. Densitometry measurements of membranes were performed to compare expression of proteins between EPCs non-OB and OB. The protein levels were normalized to β-actin /ACTB (**e**–**i**). Significant differences were calculated for normalized values and shown as means ± SD. ** *p*-value < 0.01 and *** *p*-value < 0.001, while *ns* indicates non-significant difference.

## Data Availability

https://www.biorxiv.org/content/10.1101/2021.12.02.470884v2 (accessed on 9 February 2022).

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
