# Peer review of "Obesity Affects the Proliferative Potential of Equine Endometrial Progenitor Cells and Modulates Their Molecular Phenotype Associated with Mitochondrial Metabolism"

_cells, 2022, doi:10.3390/cells11091437_

Round 1

Reviewer 1 Report

The manuscript of Smieszek et al., “Obesity affects mitochondrial metabolism and proliferative 2 potential of equine endometrial progenitor cells” describes comprehensive study of the impact of obesity to endometrial progenitor cells. Authors used the range of methods to compare numerous features of the cells derived from obese and non-obese mares and presented evidence of the differences in viability, proliferation capacity, and some metabolic changes. The study is mostly descriptive and provide the very important information about the changes in Eca EPSc (equine endometrial progenitor) cells caused by obesity. The manuscript is of great interest to the scientific community as EPS has a great potential in life science research.  

However, I have some concerns about the manuscript and suggestions to authors.

The methods that are used by authors for mitochondrial analysis are not described sufficiently. E.g. The Muse MitoPotential Kit that is claimed to be used as a kit for the characterization of mitochondrial membrane potential is rely on potential to estimate apoptosis and cell membrane permeabilization to estimate necrosis. Could the authors please elaborate more on the usage of the kit to measure mitochondrial membrane potential?

As a first step of estimation of mitochondrial health authors presented 3d reconstruction of mitochondrial network in obese and non-obese sample cells. However, it was never described which method the authors used for reconstruction and which method was used to categorize the mitochondrial shape in the cells derived from obese and non-obese mares. It would be informative to add the information of what should be expected from the different distribution of mitochondrial shapes between the samples.

Overall, all the results in mitochondrial part of study are not described thoroughly and are not followed by short summary of the obtained data and outcome. Addition of such a part would improve the quality of the results part.

The figure 6 capture states that assessment is achieved on mitochondrial net and its dynamics. Could the authors elaborate on dynamics part?

As a suggestion, the authors can consider shifting a bit a focus of the title. The manuscript describes more “molecular phenotype associated with mitochondrial metabolism” as the authors stated in the conclusion rather than mitochondrial metabolism itself.

Reviewer 2 Report

This study investigates the influence of obesity mitochondrial metabolism in equine endometrial progenitor cells. The influence of obesity on mitochondrial function is an important subject and any pertinent information is valuable.  This extensive study performed in horse cells complement other studies on the subject.  

The authors report among, many other parameters measured, altered mitochondrial dynamics and metabolism.  

  The study I thorough and I have only a few comment as follows.

1-Re senescence/proliferative potential – have any data on  the  expression of P16?

2-Fig 4, Micrographs a/b  seem not very representative compared to  the histogram c. Do you have more representative pictures?

3- fig 6, on how many cells were analyzed?

4- fig 9 what are the designation of  lanes in a?  what is ACTP (beta-actin)?  Do you have blots with lesser exposure? Maybe the blot is overexposed and quantification is not accurate? –that could perhaps explain the difference between mRNA and protein.

Discussion, it would be nice to more info on what is    known in human and rodent studies.  

Please check English language; for example row2014 “mitochondrion gene expression”

Round 2

Reviewer 1 Report

The authors addressed all my concerns in the revised version of the manuscript. The one suggestion left: add the references showing the association of elongated mitochondria with healthy cells and proper energy metabolism and globular mitochondria with overproduction of ROS and cellular falure. It will strengthen the results about mitochondrial dynamic.
